# Dog Training, Keeping and Selection around 1300, Using the Example of Albertus Magnus and Petrus de Crescentiis

**DOI:** 10.3390/ani13233698

**Published:** 2023-11-29

**Authors:** Heike Krause, Udo Ganslosser, Nina Marie Hohlfeld

**Affiliations:** 1Town Archive Gaildorf, Friedrich Schiller University Jena, 07745 Jena, Germany; 2Institute of Zoology & Evolution, Friedrich Schiller University Jena, 07745 Jena, Germany; 3Conservation Ecology Program, King Mongkut’s University of Technology Thonburi, Bangkok 10150, Thailand

**Keywords:** dog training, historical training methods, dog management

## Abstract

**Simple Summary:**

Dogs have been raised, trained, and exercised by humans for millennia. While new dog training methods keep emerging and other ones vanish, it should be interesting to travel back in time and investigate how dogs were trained throughout history. Interestingly, much of what we know about dog keeping and training has been practiced since ancient times. The oldest dog training guide was written by Athenian writer Xenophon around 420 BC. Later, in the 13th century, Albertus Magnus wrote his extensive work *De animalibus*, containing an entire chapter about dogs which covers topics ranging from behaviour and breeding to the treatment of diseases. Petrus de Crescentiis had also already reported how dogs could be used for herding and livestock protection. Using the example of these two authors, we delve into the history of ancient dog keeping and breeding.

**Abstract:**

Historical dog training methods reveal that dog training then and now might not have been that different. While some methods that would be considered unacceptable today have vanished over time, much of what we do today has been practiced historically for a long time. Albertus Magnus’ *De animalibus* and Petrus de Crescentiis’ *Ruralia commodia* deliver us historical evidence on how dogs were perceived, kept and trained by our ancestors. Not only were they already kept as pets, but they were also used in a wide range of professions. Dogs were utilized as guard and watchdogs, for hunting and for herding and livestock protection. Dogs are still trained in many of those professions today. From these historical records, we can learn how the perception and use of dogs has been similar or very different from our view on dogs today. We see how certain training methods have persisted over time, giving us an opportunity to ponder on new training and handling methods for man’s best friend.

## 1. Introduction

The number of private dog trainers or/and dog schools has increased greatly since the 1990s, and interest continues to grow. Training methods and philosophies have been developed and heralded as groundbreaking and innovative. A look at the historical literature reveals that much of what we know and teach today are methods that have been used for centuries or even millennia. On the other hand, we see that many things that would be considered inappropriate today have disappeared over time. ‘Since time immemorial’, dogs have been raised, trained and exercised by humans, as they were needed for specific tasks. Dog trainers and other professionals today often pride themselves on working with so-called “state of the art, science based methods”. However, a look into recommendations and practices from the Middle Ages already reveals a considerable amount of biological knowledge and application to practical life with dogs. “Modern” dog people thus would do better by also being aware of those earlier works before claiming “modernity”. By taking two examples from the 13th century, we want to contribute to this historical awareness. To the best of today’s knowledge, the oldest guide on the breeding, training and management of dogs was written by the Athenian writer, politician and general Xenophon (ca. 430–355 BC) and is called “Kynegetikos”. Xenophon, who was a pupil of Socrates, was the first to write a scientific treatise on the education, training and behaviour of hunting dogs, according to current research. In his work, Xenophon mentions that the gods Apollo and Artemis were accompanied by dogs on their hunts, and without them, their hunts would not have been successful.

## 2. Albertus Magnus: *De animalibus*

In the 13th century, the Dominican monk, teacher, researcher and later Bishop of Cologne Albertus Magnus (c. 1200–1280) wrote the 26-volume Latin work *De animalibus*, the exact date of which remains elusive. Research suggests 1262 or the mid-fifties of the 13th century, or after 1268 [1]. While Books I to XXI refer to Aristotle, Books XXII to XXVI were, according to Hermann Stadler, “written by Albertus himself […] with extensive use of contemporary literature” [2]. Albertus always names the sources to which he refers in his statements. “Canis”, the dog, is the subject of book XXII of the alphabetical volumes.

Albertus’ manuscripts were translated from Latin in the following decades and centuries, and printed after the invention of printing. A German edition appeared in 1545: “Thierbuch. Alberti Magni/Von Art Natur und Eygenschafft der Thierer/Als nemlich Von Vierfüßigen/Vögeln/Fyschen/Schlangen oder kriechenden Thieren/Und von den kleinen Gewürmen die man Insecta nennet/Durch Waltherum Ryff verteutscht’, printed Frankfurt am Main. The engravings and illustrations in the book are by Walter Ryff. This article refers to this edition; unfortunately, Ryff’s work does not include page numbers. As in the original Latin text, the animals dealt with are arranged alphabetically; the dog appears in “The First Book” under “C”: “Canis”.

As well as describing the nature and behaviour of the dog, the topics cover breeding, the treatment of diseases and injuries in dogs, and the use of dog excrement, milk or organs for human medical purposes. The emphasis is on training the dog for various tasks, for, as Albertus points out, the dog is “useful and entertaining” for many things, and even “the meanest little dogs” could be taught to “sit upright before the table, await their master’s kindness, and report all commotions by barking at night”. Ryff translates the Latin “exercere” from the Albertine Urtext as “gewöhnen” (to train), which had been the common term since Old High German. “Dressur”, i.e., “to train a living being”, only appears in German in the 18th century, and then only in a negative sense [3].

Albertus Magnus confines himself to three types of dogs and their tasks, namely hunting, guarding and “juggling”; there is no mention of shepherd and sheepdogs, i.e., dogs used for herding, driving and protecting livestock. This may be related to his own social context: as a member of the upper class, he had no contact with them.

### 2.1. Hounds

Hunting was originally used to provide food for people, but from the early Middle Ages, it became the exclusive prerogative of the nobility. The provision of food became an aristocratic pleasure, which was of great importance in social life and often led to conflicts with the peasants. Hunting dogs were needed for many different tasks. When the main line of Hohenlohe-Waldenburg split into the Pfedelbach, Schillingsfürst and Waldenburg lines in 1615, not only was the dominion divided, but also the hunting dogs. At that time, there were a total of 87 “Rüden”, 24 “hounds” and 13 “greyhounds” or “English hounds” [4]. “Rüden” originally meant large, strong, Molosser-like dogs, but from the late 15th century onwards, this was understood to mean “Hetzhunde” (‘hounds’) [5] that chased and attacked large game, such as bears, wolves or wild boars (see Figure 1 below). “Jagdhunde” (hunting hounds) pursued game of all kinds in packs and drove it in front of the shooter’s rifle, and lead dogs were initially dogs led by the hunter on long leashes that followed “the track of the game” [5]. In the 16th century, this developed into the pointing dog. The Swiss physician and naturalist Konrad Geßner (1516 to 1565) reported in his “Thierbuch”, the first printed book on zoology [6] (p. 218), that the “Leithund” (leader dog) was “naturally endowed with a very sharp scent”. In addition, “The hound should be completely white and completely different from the game, so that the hunter is not deceived by the similarity of the colour. […] Among the lead dogs, those are to be praised above all others who, having sniffed out the game, stand still and wait for the hunter’s approach” [6] (p. 218f). The “noblest and most beautiful of all hunting dogs”, according to Geßner, is the English greyhound: “This dog is said to have the fastest run of all hounds, is quite bold and courageous, and will not only attack the game, but also the enemy and the murderer, if he is attacked or sees that his master or handler is offended.” [6] (p. 223).

Albertus mentions the hunting dogs “Weidhund [7] or Windspiel” and “Rüden”; the latter, according to Albertus, were bred from Windspiel and Bracke. In general, the training of hunting dogs should not begin until the dog is at least two years old. Albertus’ age recommendations are quite different to what is common in hunting dog training today, where junior hunting training and tests are commonly implemented. To give just one example, Bednarek and Slawinska conducted a study in 2019 to evaluate the hunting potential of pointing dogs based on so-called junior hunting tests. The ages of the dogs in the study ranged from 9 months to 24 months [8]. This study is one of many, and it is currently a wide-spread practice to subject dogs far under the age of one year to specific training for certain types of dog sport, and even holding junior dog competitions. This definitely could be avoided by considering Albertus’ recommendations. A study by Christiansen, Bakken and Braastad in 2001 for instance found that when training hunting dogs to not attack domestic sheep, dogs under the age of three years were much more prone to undesired attacking compared to older dogs. Dogs up to the age of 3 years attacked sheep at a rate of over 50% during the entirety of their study, while older dogs only attacked sheep in around 22% of cases [9]. Albertus furthermore notes that if dogs used for hunting are physically overworked while still growing, their limbs will be damaged “while they are still soft and tender”. The result is that they are useless as hunting dogs and the dogs “become even more tired of hunting”. It is important that hunting dogs, in particular, “first become accustomed to work and fast running”. Only then will they “gain the desire and will to hunt and run fast”. It is also important that the hound is fed and that it “rests and sleeps a little” after feeding. It is fascinating to see how the concept of welfare in working dogs was already mentioned as early as the 1300s. The topic of animal welfare and handling guidelines is emphasized more and more nowadays and is addressed in many papers on working dogs. One popular approach to assess a working dog’s welfare is provided by the Five Domains model—nutrition, environment, physical health, behavioural approach and the impact on the animal’s mental state [10]. While the Five Domains model is regarded as a “modern approach” to dog keeping and training, we can see that Albertus already addressed many of these factors in his initial thoughts on dog training: nutrition, physical health, the concept of motivation and a good mental state of the dog are all mentioned by him, showcasing that even in ancient times, dogs were already perceived as sentient beings that should be positively reinforced instead of punished. Especially in modern Schutzhund training, attention is now placed on positive reinforcement versus aversive training methods, which were previously the most common methods. Nowadays, operant and classical conditioning are the methods of choice to train guard dogs [11].

### 2.2. Guard and Watchdogs

According to Albertus, not every dog is generally suitable as a watchdog to protect the house and grounds at night “against intruding thieves”. Guard dogs must be “biting” and “fierce” and “know the people”, so to speak, and defend against anyone who does not belong to the family or the household (servants, etc.). The simplest way to train a dog was to lock it up and chain it during the day and let it run free at night. Dogs kept in this way would, of their own accord, attack “everything they came across” at night, thus protecting people, house and yard “from burglary or robbery” by thieves.

Albertus also describes how a dog can be trained to attack people: you need a man dressed in a “stiff, thick, wretched skin”, meaning a protective suit made of elk leather that the dog “cannot easily tear”. The dog owner is supposed to “set his dog on him” and the person dressed in leather is supposed to run away, to flee. When the dog is released, it is supposed to jump on the man and throw him down. It is important that after the fall the person puts his face on the ground for safety, because the dog is supposed to spend some time over his “victim” “getting angry […] with biting”. These actions are carried out every day for a while. It is important, however, that the ‘victim’ is not always the same man, but someone else whom the dog does not know, so that the dog is not conditioned to a particular person. This recommendation shows interesting parallels to the Pavlovian classical conditioning, a training technique that is still widely utilized in dog training today. The fleeing man that Albertus describes, first a neutral stimulus, becomes a conditioned stimulus when the owner repeatedly sets his dog to attack him. Pavlovian conditioning is still used today, more often in the form of clicker training, for instance, when training scent-detection dogs in the police workforce to alert their handlers to a previously learned target scent [12]. We also see the training tactics mentioned by Albertus Magnus still being implemented today in police dog training: dogs are trained to apprehend suspects in a variety of circumstances and are taught to stop their target from fleeing, “to bite and hold” until their handler arrives [13,14]. They do so by biting the target on command of their handler. Police dogs are also trained on how to drag suspects out from under structures in a position that makes it safe for handlers to apprehend the target. The use of protective gear during a trained attack is still implemented in police dog training today [15].

### 2.3. ‘House Dog’/‘Domesticated Dog’

Dogs, says Albertus, regardless of their ‘species and sex’, can learn ‘almost’ anything, including tricks, or what he calls “gauckelspiel” (juggling). The simplest or most common way is for the young dog to ‘learn’ from an already trained, experienced dog. The young dog will imitate the experienced dog on his own. This is an early form of ‘do as I do’ [16]. However, Albertus stresses that only with a bitch as a ‘role model’ would a young dog become “uber die Maß gescheyd”, i.e., extremely clever; that is, only a bitch could be the ‘teacher’. What Albertus mentions here is a concept that is widely acknowledged and utilized today, which is the concept of social learning. Dogs learn not only from humans but also from their conspecifics [12]. A study by Slabbert and Rasa (1997) found that pups that received extended maternal care (3 months) performed a novel task significantly better than the pups that were separated from their mothers earlier (6 weeks). However, they conclude that the differences in performance were mostly due to differences in early observational learning and not related to extended maternal care [17]. Whether observational learning in puppies is mostly acquired through observing their mother, or if any conspecific will do, would be interesting to examine in future studies. Albertus also notes that a monkey raised and kept with the dog is suitable for learning tricks. The dog will, as it were, take on the behaviour of the monkey.

## 3. Petrus de Crescentiis: *Ruralia commoda*

Between 1304 and 1309, the Italian naturalist, advocate and later senator of Bologna, Petrus de Crescentiis (1230—c. 1320/21), wrote his major work on agriculture and botany, *Ruralia commoda* [18]. The first German translation appeared in Augsburg in 1471 [19], and another in Speyer in 1493 by Petrus Drach [18], to whom this essay refers [20]. Chapter “LXXVIII” of the ninth book is entitled: “Von hunden zu erwelen unnd leren. Unnd was ir nutz sy. unn sye zu neren”. So, it is about the selection of dogs, their training, their ‘usefulness’ and their diet. For Petrus, there are two different types of dogs: the hunting dog and the shepherd’s dog. As his work is concerned with agriculture, Petrus only describes the shepherd dog, the guardian of the animals, the sheep, goats and pigs. For this purpose, the dog was created by God to protect the cattle from the wolf and to defend them if necessary. Or, in the words of Petrus: “wydder des wolfes arge lyst”, which translates to “against the wolf’s cunning”. For herding, the dogs must not be too young or too old; otherwise, they could neither ‘overpower’ nor ‘chase’ the wolf. The stature of the dog is also important; it should be ‘beautiful and large’, ‘with black eyes, grey or red lips’, a large head, drooping ears, a ‘thick’ neck, a long tail and ‘a rough voice, strong breath to run’. The coat colour should ideally match that of a lion. It is always important in the keeping of working dogs to feed extensively, for a hungry dog will ‘hunt for its food’. Shepherds must never buy their dogs from tanners, as they would not have learned to follow the cattle. A dog bred by a hunter is also unsuitable; the risk of the dog developing a strong urge to hunt due to its background is too great. The shepherd could only buy reliable herding dogs from shepherds. Petrus sums up by saying that ‘what a dog gets used to [learns] when it is young, it will keep’. The training of herding dogs should start early, and the young dogs should be introduced to the cattle by their mothers, i.e., the young dogs ‘learn’ herding and guarding behaviour from their mothers. These statements prove that dogs were not usually given away as puppies, but only when they were young. This was common practice even in the 19th century, as evidenced by numerous newspaper advertisements. Recent research on livestock-guarding dogs has been reviewed and summarised in the literature by Coppinger’s group [21,22] and shows some remarkable similarities to Petrus’ recommendations. Coppinger and Feinstein emphasise that the traditional working methods of livestock-guarding dogs cannot be achieved through training, operant learning or other methods of refining dog behaviour. They need to be socialised with, e.g., sheep at a very early age (usually around 5 weeks) and therefore tend to regard their protegees as extended family later on. In addition, livestock guarding dogs usually work by disrupting the predator’s prey-catching sequence by showing greeting, playful and sometimes aggressive behaviour towards the predator. Thus, the effects of early socialisation, the use of the dog’s own species-typical behavioural patterns, combined with a delayed behavioural ontogeny, seem to be the reason for their successful work. Shepherds tend to observe the dogs at work, adopting those that perform well and rejecting or even culling the others.

In modern herding dog training, it is not so common that puppies will be introduced to livestock by their mothers, but they will rather be trained by their owners. However, in the selection of herding dogs today, the heritability of so-called herding behaviour is still emphasized a lot, and herding dogs are typically selected due to certain behavioural traits they must possess, many of them believed to be inherited [23].

Petrus said that it was important to get the dogs used to the leash. Petrus emphasized that the dogs should never be allowed to gnaw on the lead. Once they become used to this bad habit, they will continue to do so in their ‘old age’ or, in the worst case, they will not allow themselves to be leashed at all. To protect them from wolf attacks, sheepdogs should wear wide nailed collars. The number of dogs must be adapted to the size of the flock. Several dogs are also needed if the flocks graze mainly near woodland, where the risk of wolf attack is greater. In principle, however, at least two dogs are always needed, and always a male and a female: ‘Because then they are much more industrious and one makes the other more cheerful’. Even if one of them is sick and cannot work, the other can take over. Let them watch over your sheep at night and give them time to sleep or rest during the day.

Georg Wilhelm Friedrich Hegel (1770–1831) once said: “But what experience and history teach us is that nations and governments have never learnt anything from history and have never acted in accordance with the lessons that could have been drawn from it”. This statement also applies to dog training and ownership in the sense that the lessons of past centuries are being lost. We can learn from history what is positive and worth continuing and developing, but we should avoid the negative. Albertus’ reference to the female dog and Petrus’s reference to the mother dog as a ‘teacher’ could, for example, encourage sex-specific research in behavioural biology to verify this statement in relation to herding.

## Figures and Tables

**Figure 1 animals-13-03698-f001:**
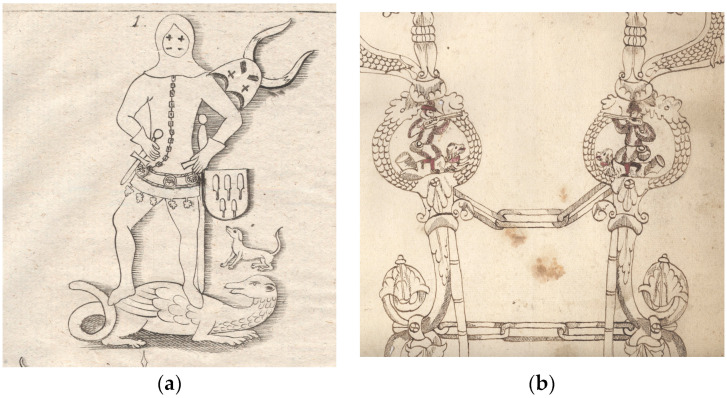
(**a**) Engraving of an epitaph for Albrecht I von Limpurg (+1374) in the Schenkenkapelle of Comburg Abbey in Schwäbisch Hall (from: Heinrich Prescher: Geschichte und Beschreibung der […] Reichsgrafschaft Limpurg, Volume 1, 1789); (**b**) Drawing of a horse bridle (detail), c. 1600 (Gaildorf town archive, inv. no. OBA01/978).

## Data Availability

No new data were created or analysed in this study. Data sharing is not applicable to this article.

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
