# Peer review of "Dog Training, Keeping and Selection around 1300, Using the Example of Albertus Magnus and Petrus de Crescentiis"

_animals, 2023, doi:10.3390/ani13233698_

Round 1

Reviewer 1 Report

Comments and Suggestions for Authors

I enjoyed this interesting historical Commentary on early dog keeping, training, and working roles. My main concern is that after reading the Simple Summary and Abstract, I expected to see some direct comparisons in the text of early training methods with current training techniques (e.g., which methods are still used today, and which techniques have been modified or dropped?). I realize this paper is not meant to review current training techniques, but it would be useful to broaden the scope a bit and reference some general similarities and differences between methods around 1300 and today. At the end of this review, I include some relatively recent papers on current dog training methods that might be helpful.

The Commentary also seemed to end abruptly, without any concluding remarks. 

Some minor issues are detailed below:

Line 20: Change “this” to “that”

Lines 33-34: It would be good to include a citation at the end of this first sentence.

Line 82: What is meant by “moss-like dogs”?

Lines 99-100: Here is a place where you could compare past and present practices – for example, are dogs currently trained to hunt once they are two years or older?

Lines 115-124: Here is another place where you could generally compare the training of guard dogs in the 13th century to that of police dogs and guard dogs today.

Lines 129-131: Is there any current evidence that young dogs learn better from female dogs than male dogs?

Line 145: it would be helpful to readers if you provided the translation of “wydder des wolfes arge lyst"

Lines 155-157: How are dogs trained in herding today – are the methods similar or different from those described by Petrus?

Potentially helpful papers on training techniques today:

Hall NJ, Johnston AM, Bray EE, Otto CM, MacLean EL and Udell MAR (2021) Working dog training for the twenty-first century. Front. Vet. Sci. 8:646022. doi: 10.3389/fvets.2021.646022

China L, Mills DS and Cooper JJ (2020) Efficacy of dog training with and without remote electronic collars vs. a focus on positive reinforcement. Front. Vet. Sci. 7:508. doi: 10.3389/fvets.2020.00508

Vieira de Castro AC, Fuchs D, Morello GM, Pastur S, de Sousa L, Olsson IAS (2020) Does training method matter? Evidence for the negative impact of aversive-based methods on companion dog welfare. PLoS ONE 15(12): e0225023. doi: 10.1371/journal.pone.0225023

Vieira de Castro AC, Araujo, Aˆ , Fonseca A, Olsson IAS (2021) Improving dog training methods: Efficacy and efficiency of reward and mixed training methods. PLoS ONE 16(2): e0247321. doi: 10.1371/journal.pone.0247321

Ziv, G. 2017. The effects of using aversive training methods in dogs - A review.  Journal of Veterinary Behavior 19: 50-60. doi:10.1016/j.jveb.2017.02.004

Author Response

Please see the attachment for our response to your comments.

Reviewer 2 Report

Comments and Suggestions for Authors

This manuscript provides interesting details of the writings on dogs by two 13th Century authors that have not received much attention in the literature of dog training and keeping. However, the abstract is disappointingly misleading. It suggests that “we can learn how the perception and use of dogs has been similar or very different from our view of dogs today” and “we can see how certain training methods have persisted over time.”

Despite this promise - there is virtually no discussion of contemporary dog training or the ways in which it is like or unlike that described by Albertus Magnus or Petrus de Cresentis. In fact, the only mention of a modern dog training ideas is the passing reference to Topal et al. in the note about the idea of using experienced dogs for “do as I do” training. There could have been much more discussion of how this concept is reflected in modern approaches to training of guarding, livestock or detection dogs.

Most of the references to the 13th Century work focuses on selection and keeping rather than training, so the title itself is misleading. Yet there is no discussion of how these views on selection conform to or differ from modern ideas.

The abstract promises “giving us inspiration for new training and handling methods…”, but that promise is not even addressed. There are many contemporary issues in modern day dog training that might benefit from the ideas from centuries ago -  for example “what is the best way to reward desired behavior (food, praise, play?)”; “Does punishment work, and what form should it take?”; “How should training be scheduled - at what age, on what schedule”

Comments on the Quality of English Language

A few typos and punctuation errors

Author Response

Please see the attachment for our reply to your comments.

Round 2

Reviewer 2 Report

Comments and Suggestions for Authors

The revised manuscript is much improved and has addressed most of the issues raised in the initial critique. It would still benefit from a few minor additions and edits.

The mention of Xenophon’s “Kynegetikos” (line 47) would be helped by some additional commentary about its content and whether it was available to Albertus Magnus or Petrus de Cresentiis.

The commentary on Guard and Watchdogs (line 155 on) should go beyond simply mentioning Pavlovian conditioning. There is extensive literature on plocies and procedures in training police and guard dogs, much of which parallels methods mentioned by Albertus Magnus, such as the use of protective sleeves.  An overview of some of this is available through the US Police Canine Association (uspcak9.com/k9-training-articles). In addition, there is extensive literature on guard dog training, particularly for Shutzhund and Ring sports. Shutzhund training has undergone major changes in the last 50 years, moving away from more aggressive - punishment based approaches, which again might bring them more in line with these older recommendations. There is a wealth of information on modern Shutzhund training available online.

The manuscript would also be improved with greater attention to modern views of livestock guarding dogs, since that was a significant concern of the early writers. The authors do not mention any of the modern literature on the topic, such as the extensive work of Dr. Raymond Coppinger on livestock guarding dogs, including Coppinger (1978). Livestock guarding dogs for U.S. agriculture.

Comments on the Quality of English Language

Some of the grammar and sentence structure is a bit stilted in translation requiring some light copy-editing.
